# One Step Back or One Step Forward? Effects of Grade Retention and School Retention Composition on Portuguese Students' Psychosocial Outcomes Using PISA 2018 Data

Joana Pipa * and Francisco Peixoto 

CIE-ISPA (Centre for Educational Research-ISPA), ISPA-Instituto Universtiário, Rua Jardim do Tabaco 34, 1149-041 Lisbon, Portugal
* Correspondence: jpipa@ispa.pt

**Abstract:** Grade retention is a common practice applied to academically struggling students within the Portuguese context. Studies investigating the psychological experiences of grade-retained students are still scarce. In addition, most studies tend to neglect the multilevel nature of the school context. This study examines the effects of grade retention in grades 1–9 on Portuguese students' psychosocial outcomes by the age of 15, using PISA 2018 data. Using a quasi-experimental design through full matching, we reduced the bias between 1362 retained and 4189 promoted students in relevant background variables. Results from the multilevel models showed that retained students, by the age of 15, present lower task orientation and school belonging. In addition, we found that the high retention rates negatively relate to students' reading self-concept, task orientation, and school valuing and that school retention rates moderate the relationship between students' retention and the psychosocial variables considered. Overall, these findings suggest detrimental effects of grade retention and that grade retention also affects the promoted peers of retained students.

**Keywords:** grade retention; psychosocial outcomes; school retention composition; quasi-experimental methods; multilevel modelling; PISA

## 1. Introduction

When students struggle to meet the academic expectancies or goals established for a specific grade level, one option is to allow them more time by retaining them in the same grade [1,2]. This type of response to deal with students' heterogeneity, in terms of academic competencies, is one of the most discussed educational strategies [3]. Furthermore, each country usually uses grade retention rates as a measure of educational quality and equity [4–6].

Accompanying the long debate around grade retention effectiveness, most countries decreased their retention rates during the last decade. However, in countries such as Belgium, Luxembourg, Germany, Spain, and Portugal [6], grade retention is still a common strategy used to deal with students' low achievement [5,7–9]. This paper focuses on data from Portugal, where grade retention rates are among the highest in the Organization for Economic Co-operation and Development (OECD) countries. In these educational systems, grade retention is believed to bring several benefits to struggling students, such as giving them more time to develop and mature and to catch up on the learning materials [5,10,11]. In addition, retaining students in grades is believed to improve teacher effectiveness by creating more homogenous classrooms, and the threat of being retained might boost students' motivation to work harder [7,10,12].

However, opponents of grade retention argue that grade retention harms students' motivation, confidence, and sense of self-worth because, as they will go over the same curriculum once more, they are deprived of intellectual challenges and meaningful learning [13]. Moreover, retained students are detached from their peer group and will face a

new classroom of younger students, leading to a decreased perceived competence and a sense of failure [10]. Finally, being retained brings extra economic and opportunity costs for the educational system, students, and families [11,14].

Being retained could constitute a rather negative psychosocial experience for students [15–17] and has been pointed out by children and adolescents as one of the most stressful life events [15], affecting their motivation and self-confidence [10,13,17–19]. Nevertheless, despite its relevance for educational success and adjustment, studies considering the effects of grade retention on students' psychosocial outcomes, such as self-concept, motivation, or engagement, have received less attention from the research community [20]. In addition, as advised by other scholars, school context plays a crucial role in shaping students' self-beliefs and experiences [9]. In particular, school retention composition, i.e., the proportion of repeaters in a given school, was associated with academic and psychosocial outcomes, namely, students' peer relationships, self-concept, school belonging, and enrolment in post-secondary education [21–24].

Therefore, this paper investigated whether Portuguese students with a history of grade retention differ from their promoted same-age peers in psychosocial outcomes. Moreover, we aimed to study whether these differences could also be exacerbated or not in schools with a more significant proportion of retained students. For this study, we relied on the available data from the Programme for International Study Assessment (PISA) 2018 regarding the Portuguese context.

### 1.1. Previous Research on the Effects of Grade Retention

Research on grade retention effectiveness has grown tremendously, especially since 2010 [20]. Recent meta-analyses and systematic reviews estimated an overall null effect of grade retention [1,20,25]. However, these studies found that the impact of grade retention is highly dependent on the context where it occurs (e.g., country, state, and educational system), the timing of grade retention, the timing of follow-up (e.g., short-term studies vs. long-term studies), and the variables observed (e.g., academic achievement vs. psychosocial or school career) [20,25].

The existing studies considering psychosocial outcomes are far less conclusive than those investigating academic achievement outcomes, suggesting positive, negative, or nonsignificant effects [3,20,25]. These inconclusive findings could be attributed to the broader nature of the term 'psychosocial outcomes', covering different variables that could be differently affected by grade retention [20,22]. Additionally, short-term studies tend to present more positive results [10,14,18,22,26,27].

#### 1.1.1. Effects on Students' Academic Self-Concept

Students' academic self-concept, defined as students' self-perception of competence in specific academic-related domains (e.g., reading self-concept), plays a significant role in school adjustment, achievement, and educational success [9]. To explain the effects of grade retention on students' academic self-concept, researchers have often referred to the big-fish–little-pond effect [28]. This effect posits that students compare their own school-related accomplishments with those of their classmates, and this frame of reference act as the base for their self-concept development [9,28,29].

Based on this framework, one could expect that retained students would develop more positive academic self-beliefs, at least during the retention year, because their frame of reference would comprise their younger grade mates with less academic experience [9,29]. On the other hand, however, some scholars also claim that being retained jeopardises students' perception of competence because they may perceive that being retained constitutes a personal failure that makes them less competent and capable [3,11].

Thus, the empirical evidence also shows mixed effects of grade retention. Longitudinal studies assessing grade retention in primary [18] and lower secondary education [14,18,29–31] found either positive effects on math and academic self-concept [14,29,30], or adverse effects on language, math, and academic self-concept [18,31], during the retention year. Conversely,

more adverse long-run findings emerged from cross-sectional studies investigating effects in lower [26] and upper secondary education [17,22]. In contrast, longitudinal studies generally revealed nonsignificant effects on students' academic, language, and math self-concept during lower and upper secondary education [26,29–32]. Positive long-run effects were found in only one study using international PISA data [9].

### 1.1.2. Effects on Students' Goal Orientations

Students' motivation has received much attention from researchers as it is recognised for its critical role in students' academic behaviour and performance [33]. Students' goal orientations have become one of the largest research fields in motivation and are characterised by students' reasons or purposes for engaging in certain achievement behaviours [34]. The PISA in 2018 assessed two kinds of these reasons, or orientations, to engage in academic tasks: task orientations and self-enhancing ego orientations [35]. In task orientation, students engage in a school-related task (e.g., a reading task) to develop and acquire knowledge or master a new skill. On the other hand, when students pursue a self-enhancing ego orientation, the aim is to demonstrate competence and outperform others [35,36].

Considering the effects of grade retention on motivational outcomes in general, one may expect, on the one hand, that grade retention acts as a 'boost' in students' motivation since they will finally experience success and will receive positive feedback from teachers [10]. Additionally, students may view the retention year as a second chance to master the learning content or even an opportunity for a fresh start [31,37].

Despite its relevance within the educational context, to our knowledge, studies investigating the effects of grade retention on students' goal orientations are still scarce. The existing studies mainly suggest that students with a retention history demonstrate less adaptive motivational profiles in lower secondary education, even before being retained [31], and during the retention year and beyond [19].

### 1.1.3. Effects on Students' Sense of School Belonging and Valuing

Apart from students' individual motivation, their social exchanges within the school context, particularly with their peers, teachers, and the broader school community, are essential agents in shaping their motivation at school [38]. This social aspect of school motivation is commonly known as students' sense of school belonging [38–41]. Students' sense of school belonging can be understood as students' feelings of being accepted, respected, valued, and supported by their peers and the broader school community [38–40]. In addition, these feelings of belonging are also associated with valuing school and school success, the two components of students' participation at school [39].

When students are retained in a grade, they lose their peer group. This experience of a broken relationship can trigger feelings of isolation and alienation from school [10,21] due to failure to satisfy the need to establish and, especially, to maintain stable relationships with others [41,42]. In addition, grade retention is an explicit form of academic failure. This stigma of failing a grade and not being good enough academically can make establishing new and positive relationships in a new and unfamiliar peer group even harder [21]. These feelings and experiences may thus lead to a greater sense of school disaffection and feelings of being an outsider from school and not connected with the school community [10,21].

Empirical studies investigating the effects of grade retention on students' sense of belonging suggest that grade retention does not improve students' sense of school belonging. Longitudinal studies showed predominantly adverse effects in both primary [27] and lower secondary education [43]. Cross-sectional studies revealed mainly adverse effects of retention on school belonging [22], particularly those using international PISA data [23,44]. To our knowledge, the effects of grade retention on students' school valuing as it is operationalised here are inexistent.

### 1.2. School Retention Composition

Empirical studies recognising the importance of school context in studying grade retention, although less common, have been growing during the last years, suggesting its crucial role in moderating the relationship between individual grade retention and academic and psychosocial outcomes [12,21–24,32,45–47]. In addition, Van Canegem et al. [22] posited that the context where grade retention occurs might be crucial to clarify divergent findings from previous studies.

The number of retained students attending a particular school can impact students' psychosocial outcomes in two ways. First, a direct effect of school retention composition is expected through the so-called spillover effects of retained students on their non-retained peers [22]. Spillover effects of grade retention have been less considered in grade retention research, despite constituting a big concern for families and educators [20]. Retaining students in a particular classroom may negatively affect the classroom climate and instruction and, therefore, the learning of non-retained classmates [24,48].

Second, school retention composition may moderate the relationship between individual grade retention and students' psychosocial outcomes. Accordingly, retained students might present more negative behaviours and feelings of being stigmatised in schools with low rates of grade retention. These feelings might, thus, exacerbate the impact of grade retention on students' self-concept and sense of belonging, for example [21–23,45].

The empirical studies that have addressed the impact of school retention composition on students' psychosocial factors showed that students from schools with many repeaters tend to be more likely to misbehave at school have a lower number of friends, and lower levels of academic self-concept and school belonging [21–23,45]. In addition, some studies showed more favourable results of grade retention in schools with higher retention rates [21–23].

### 1.3. The Portuguese Context

The Portuguese school system, along with other southern European countries (e.g., France, Italy, and Spain), offers a common core curriculum for all students until 9th grade, and grade retention is the primary strategy applied to deal with academically struggling students and to promote homogeneity inside the classroom [7]. In Portugal, 24% of 15-year-old students have reported being retained at least once during their school career, being largely above the OECD average of 11% [6]. These rates may reflect the 'culture of retention' mentioned in several studies, meaning that educators believe that grade retention is beneficial for students over and above the recommendations of the international educational community and even national legislation [5,8]. Currently, the national legislation states that grade retention in Portugal should only be an 'exceptional measure' when promoting the student to the next grade compromises the acquisition of new learnings [49]. Thus, grade retention decision falls on schools and teachers who, except for transition years, have the autonomy to define in which specific circumstances grade retention will be applied.

Although it is a widespread practice in Portugal, studies using Portuguese data are still scarce, especially in using adequate methodologies and considering variables beyond student achievement [50,51]. To our knowledge, only a limited number of studies have focused on the effects of grade retention on students' psychosocial outcomes, such as students' self-concept and motivation. These studies showed predominantly adverse effects [18,19,47].

### 1.4. The Present Study

In summary, the following considerations guided this study: (a) the findings that the characteristics of each school system moderate the effects of grade retention; (b) the limited number of empirical studies evaluating grade retention effects within the Portuguese context (and to some extent considering other countries with a similar educational system); (c) the mixed empirical evidence of the effects of grade retention on students' psychosocial

variables driven by the differential nature of each construct; and (d) the importance of school retention composition in clarifying the relationship between grade retention and academic outcomes.

Hence, we aimed to investigate the effects of grade retention on a group of psychosocial variables. Specifically, using a large-scale assessment and applying a same-age comparison approach, we examined (1) whether students who had experienced grade retention at least once between grade 1 and grade 9 differed from their same-age promoted peers in reading self-concept, goal orientations, and students' sense of school belonging and valuing; and (2) whether the nature of these effects can differ according to the proportion of retained students attending a school, i.e., school retention composition.

## 2. Materials and Methods

### 2.1. Data and Participants

Starting in 2000, PISA became one of the largest and most prominent large-scale assessment studies in education. Every three years, PISA assesses reading, math, and science competencies acquired by 15-year-old students. In addition, PISA gathers information regarding student and family background information and various psychosocial variables, including academic self-concept, motivation, and school engagement.

This study used data from 5932 Portuguese students who participated in PISA 2018, attending 276 schools. Since this study focused on grade retention effects, we excluded students who had missing information (5%, $n = 308$) in the grade retention variable from the data. Thus, further analyses were based on 1362 retained (24%) and 4262 promoted students ($M_{age} = 15.73$, $SD_{age} = 0.29$, 50% boys), attending between 7th and 11th grade. PISA uses a two-stage process regarding the sampling procedure to obtain a representative sample of students from each country and economy. First, schools are randomly selected from a complete list of schools containing the student population of interest. Second, 35 15-year-old students from each school are randomly selected to fulfil the questionnaires [52].

### 2.2. Measures

In this study, we focused on PISA data considering students' reading self-concept, goal orientations, and school belonging and valuing. In addition, we used information regarding students' social background and school characteristics retrieved from students' and school principals' questionnaires. All continuous measures were standardised to have a mean of 0 and a standard deviation of 1, and categorial measures were coded as dummy variables to facilitate interpretation. Concerning the validity and reliability of the measures used, internal consistency ranged from $\alpha = 0.74$ to $\alpha = 0.88$ (see Table 1), revealing acceptable levels of reliability, as was also referred to in the PISA report [44].

**Table 1.** Means and standard deviations for retained and promoted students, internal consistency, and correlations between the outcome variables.

| | Repeaters | | Non-Repeaters | | Internal Consistency | | | | | |
|---|---|---|---|---|---|---|---|---|---|---|
| | *M* | *SD* | *M* | *SD* | $\alpha$ | 1 | 2 | 3 | 4 | 5 |
| 1. Reading self-concept | −0.50 | 0.83 | −0.16 | 0.83 | 0.74 | - | | | | |
| 2. Task orientation | −0.33 | 0.99 | 0.11 | 0.93 | 0.83 | 0.25 *** | - | | | |
| 3. Self-enhancing orientation | −0.04 | 0.96 | −0.06 | 1.00 | 0.87 | 0.06 *** | 0.08 *** | - | | |
| 4. School belonging | −0.02 | 0.97 | 0.17 | 0.97 | 0.80 | 0.13 *** | 0.17 *** | 0.10 *** | - | |
| 5. School utility value | 0.09 | 0.97 | 0.50 | 0.85 | 0.88 | 0.14 *** | 0.32 *** | 0.01 | 0.19 *** | - |

*** $p < 0.001$.

### 2.2.1. Grade Retention

In the PISA questionnaire, students were asked whether they had ever repeated a grade in ISCED I, II, or III. This study considered grade retention responses regarding ISCED I or II. In these variables, 0 means that a student never repeated a grade during ISCED I or II, whereas 1 means that a student has repeated it at least once. In addition,

school retention composition was derived from responses to grade retention in ISCED II and III variables since many schools offer both these two levels of education and were operationalised as the percentage of retained students in each school.

### 2.2.2. Reading Self-Concept

To assess academic self-concept in reading, students were asked on a 4-point scale (strongly disagree to strongly agree) whether they perceive themselves as good readers, whether they are able to understand complex texts, and whether they read fluently [53].

### 2.2.3. Goal Orientations

Students' task orientations, or learning goals as mentioned by PISA [53] (p. 215), were measured on a 5-point scale (not at all true of me to extremely true of me), asking to what extent they have the goal of learning and master class-related materials. Likewise, to assess self-enhancing orientations or attitudes toward competition, as referred to by PISA [53] (p. 215), students answered three items on a 4-point scale (strongly disagree to strongly agree), asking whether they enjoy working in competitive environments and whether they have the goal of outperforming others.

### 2.2.4. School Belonging and Valuing

To assess their sense of school belonging, students answered six statements, such as 'I feel like an outsider (or left out of things) at school' and 'I feel like I belong at school' [44] (p. 130), on a 4-point scale (strongly disagree to strongly agree). Students also completed three items to assess how much they value school. In these statements, students were asked whether they agree (strongly disagree to strongly agree) that trying hard at school would help them obtain a good job or help them be accepted into a good college [44].

### 2.2.5. Students' Social Background and Competencies

We considered several individual background variables retrieved from the students' questionnaire and the measurement of competencies that were related to grade retention [9,14,50,54–57]. Specifically, we retrieved information considering students' age, gender, immigrant background, language spoken at home, index of economic, social, and cultural status (derived from parents' highest level of education, parents' highest occupational status, and home possessions), home educational resources (i.e., household possessions and the number of books at home), and parents' emotional support (self-report measure where students where asked whether they feel supported by their parents) [53]. In addition, we considered students' PISA scores in reading, math, and science.

### 2.2.6. School Context

We also integrated some school-related components as covariates considered in the literature related to grade retention [24,54–57]. Specifically, we retrieved information regarding school type (public vs. private) [53]. The proportion of participating students attending public schools does not differ from the true proportion of Portuguese students attending such schools (88% in both cases) [4]. In addition, school composition in terms of the index of economic, social, and cultural status and immigrant background, obtained by aggregating students' responses, was also considered.

### 2.3. Data Analysis

### 2.3.1. Handling Missing Values

In most observational studies, participants often leave one or more questions unanswered. Researchers have been encouraged to deal with incomplete datasets in recent years by applying imputation methods. In this study, we applied multiple imputations by chained equations using the MICE package in R [58], generating five completed datasets and allowing ten iterations. The proportion of missing values in the variables used in the

analyses ranged from 0.1% to 10%. Subsequent analyses were conducted in each imputed dataset and then aggregated [59].

2.3.2. Group Comparison Strategy

Applying an experimental methodology to estimate the causal effects of grade retention is not theoretically or ethically attainable, as students cannot be randomly retained or promoted. Nevertheless, the literature on grade retention effects draws attention to the importance of establishing comparability between the intervention (i.e., retained students) and comparison (i.e., promoted students) groups [1,20]. Thus, propensity score matching methods are often applied to reduce selection bias by achieving a balance between treatment and comparison groups regarding background characteristics related to both the treatment and the outcome [60,61]. These methods have also been widely used in international studies, such as those using the PISA data [61]. We opted to apply the full matching technique among the different matching methods, using the MatchIt package in R [62]. The full matching technique is considered a more sophisticated and flexible matching method and has the advantage of not discarding any observation (as does, for example, the one-to-one nearest neighbour technique). Full matching forms a series of matched sets (subclasses) containing at least one treated and one comparison subject [63]. After creating these matched sets, each comparison individual receives a weight proportional to the number of treatment individuals present in each set [62]. In subsequent outcome analyses, these weights are introduced in weighted regression models [62]. Full matching techniques are recognised to be efficient in maximising the similarities between treatment and comparison individuals in each matched set [63] and have been successfully used in grade retention effects research [31]. The final step of preparing comparison groups for the outcome analysis was assessing the balance between treated and comparison subjects across the covariates used [63]. This assessment was performed considering [62,63] (a) standardised mean differences below 0.25; (b) variance ratios between 0 and 2; and (c) graphical inspection. In this study, we include in the propensity score matching a series of background characteristics and interaction terms, described in Section 2.2.5. The selection of these variables was theoretically based, as mentioned previously.

2.3.3. Outcome Analysis

The effects of grade retention on psychosocial components were estimated using multilevel models since students are nested within schools [9]. Thus, a series of hierarchical linear models were computed for each outcome, considering four stages (Model 0 to Model 3). Model 0 or the 'null model' was estimated with no predictors to examine the amount of variance in the outcomes that is explained by either student or school levels. Model 1 was estimated by entering students' background variables, reading, math, and science scores, school context variables, and grade retention variables. In Model 2, school retention composition was added to the previous variables. Model 3 assessed the interaction between grade retention and school retention composition. In estimating these models, we followed Stuart's [63] recommendations of combining matching and regression methods by including in the regression models the predictors and controlling variables previously considered in the matching procedures. We used weighted regression models, as already mentioned, considering the weights obtained after matching. Finally, in every model, we additionally checked for multicollinearity using the variance inflation factor (VIF). Using a threshold of 10 indicating a strong correlation between the independent variables, our results supported its independence, with VIF values ranging between 1.02 and 4.84.

## 3. Results

### 3.1. Covariate Balance across Retained and Promoted Students

Before conducting the outcome analysis, we assessed whether the selection bias on background variables was reduced through full matching. It is first worth mentioning that we observed extreme weights in some comparison individuals, suggesting difficulties in

finding suitable matching individuals for these observations [63]. Hence, we discarded the observations with extreme weights above the 99th percentile (*n* = 73). Further analyses were based on 1362 retained students and 4189 promoted students.

Results from the full matching revealed adequate balance. As shown in Figure 1, the absolute standardised mean differences between retained and promoted students in the background variables decreased considerably after matching, with all standardised mean differences below 0.25. In critical, highly related to retention, background variables, such as students' ESCS, the standardised mean differences were reduced to 0.01 standard deviations.

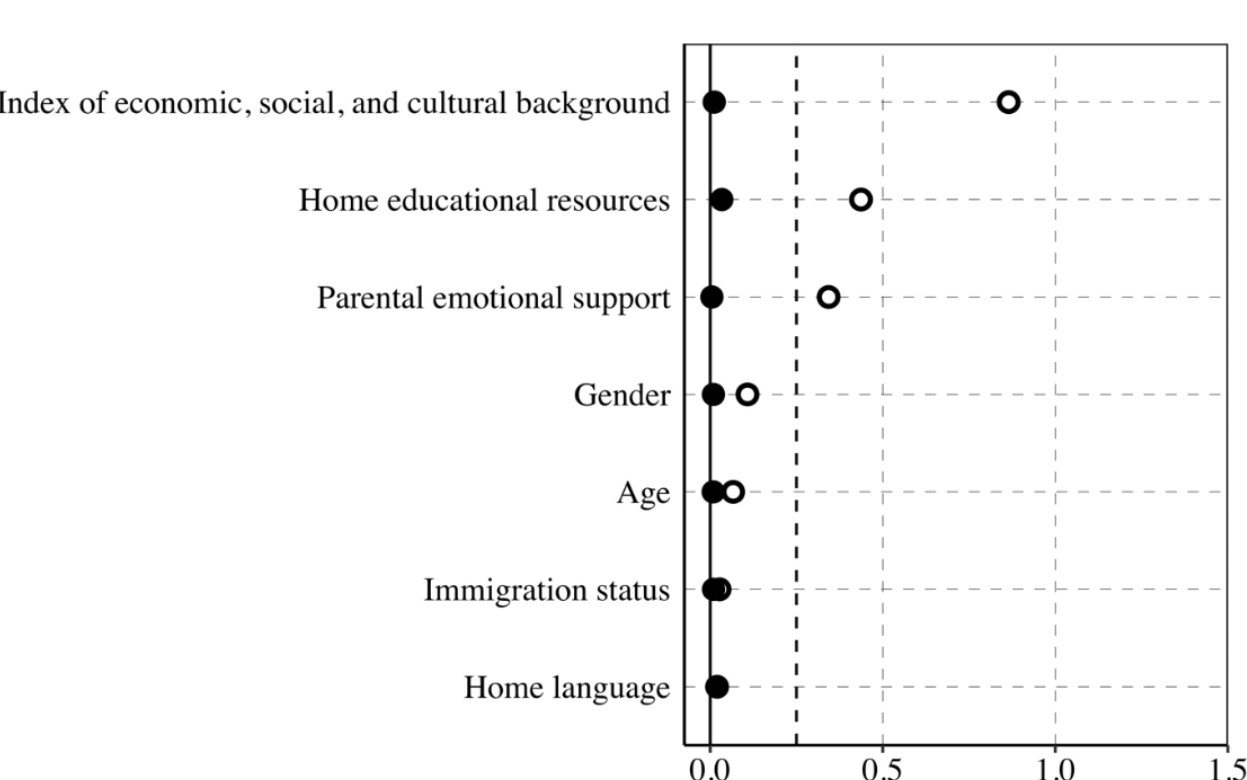

**Figure 1.** Absolute standardised mean differences between retained (*n* = 1362) and promoted students (*n* = 4189) before and after full matching.

### 3.2. Effects of Grade Retention

This section presents the effects of grade retention on the outcome variables assessed. Table 1 shows descriptive statistics in the outcome variables for the retained and promoted groups and correlations between outcome variables.

To improve readability, in Table 2, we present only the results from Model 3, i.e., the model testing both single effects of grade retention and school retention composition and interaction effects between these variables. In the following sections, we briefly describe the findings from the remaining models. Detailed information regarding coefficients from Model 0 to Model 2 for each outcome variable can be found in Appendix A.

#### 3.2.1. Effects of Grade Retention on Reading Self-Concept

The intraclass correlation coefficient (ICC) obtained from Model 0 (these coefficients are only described in the text; calculations are possible from the tables in Appendix A) suggests substantial variance between schools in reading self-concept (ICC = 0.20). Thus, these results support the multilevel analysis. Considering the main effect of grade retention,

the results were consistent across the three models; the relationship between grade retention and reading self-concept was nonsignificant. Conversely, Model 3 revealed a negative relationship between school retention composition and reading self-concept (b = −0.07, $p < 0.05$), suggesting that students in schools with a larger proportion of repeaters tend to feel less competent in reading.

**Table 2.** Multilevel unstandardised parameter estimates from Model 3 for the outcomes assessed.

|  | Reading Self-Concept | Task Orientation | Self-Enhancing Orientation | School Belonging | School Utility Value |
|---|---|---|---|---|---|
| Intercept | −0.10 (0.08) | 0.12 (0.10) | −0.05 (0.10) | 0.45 (0.10) *** | 0.49 (0.10) *** |
| Student level |  |  |  |  |  |
| Retention | 0.01 (0.03) | −0.16 (0.03) *** | −0.07 (0.03) | −0.12 (0.03) *** | −0.08 (0.03) * |
| Age | 0.01 (0.01) | 0.02 (0.01) | 0.01 (0.01) | −0.03 (0.01) ** | −0.06 (0.01) *** |
| Male | −0.06 (0.02) * | −0.21 (0.03) *** | 0.42 (0.03) *** | 0.20 (0.03) *** | −0.16 (0.03) *** |
| ESCS [a] | 0.08 (0.01) *** | 0.02 (0.02) | 0.02 (0.02) | 0.06 (0.02) *** | 0.02 (0.02) |
| Immigration status | 0.03 (0.04) | 0.20 (0.05) *** | −0.14 (0.06) * | −0.18 (0.05) ** | 0.09 (0.05) |
| Home language | −0.26 (0.06) *** | 0.05 (0.07) | −0.17 (0.07) * | 0.01 (0.07) | 0.06 (0.07) |
| HEDRES [b] | 0.10 (0.01) *** | 0.16 (0.01) *** | 0.04 (0.02) * | 0.09 (0.01) *** | 0.11 (0.01) *** |
| EMOSUPP [c] | 0.07 (0.01) *** | 0.21 (0.01) *** | 0.11 (0.01) *** | 0.23 (0.01) *** | 0.22 (0.01) *** |
| Reading score | 0.37 (0.03) *** | −0.10 (0.04) ** | −0.14 (0.04) *** | 0.10 (0.03) ** | 0.24 (0.03) *** |
| Math score | −0.31 (0.03) *** | −0.07 (0.04) | −0.21 (0.04) *** | −0.30 (0.03) *** | 0.03 (0.03) |
| Science score | 0.12 (0.03) *** | 0.20 (0.04) *** | 0.34 (0.04) *** | 0.29 (0.04) *** | −0.10 (0.04) ** |
| School level |  |  |  |  |  |
| Retention composition | −0.07 (0.03) * | −0.13 (0.04) *** | −0.01 (0.03) | −0.05 (0.03) | −0.12 (0.03) *** |
| Public school | 0.11 (0.07) | −0.07 (0.08) | −0.01 (0.08) | −0.32 (0.08) *** | −0.03 (0.07) |
| School ESCS [a] composition | −0.04 (0.03) | −0.04 (0.03) | −0.01 (0.03) | 0.02 (0.03) | 0.01 (0.03) |
| School immigrant composition | −0.01 (0.02) | 0.01 (0.02) | −0.01 (0.02) | 0.02 (0.02) | 0.01 (0.02) |
| Retention X retention composition | 0.08 (0.02) *** | 0.13 (0.03) *** | 0.03 (0.03) | 0.09 (0.03) ** | 0.14 (0.03) *** |
| Between school variance | 0.08 (0.28) | 0.10 (0.32) | 0.10 (0.31) | 0.10 (0.32) | 0.09 (0.30) |
| Within school variance | 0.31 (0.56) | 0.47 (0.68) | 0.51 (0.72) | 0.43 (0.65) | 0.43 (0.66) |
| $R^2$ | 0.21 | 0.20 | 0.12 | 0.25 | 0.25 |

Note: Standard errors are in parenthesis. [a] Index of economic, social, and cultural background; [b] Home educational resources; [c] Parental emotional support. * $p < 0.05$, ** $p < 0.01$, *** $p < 0.001$.

In addition, the results showed a combined effect of grade retention and school retention composition (b = 0.08, $p < 0.001$) on this outcome, revealing that the higher proportion of repeaters in school attenuates the effects of grade retention on students' reading self-concept. Figure 2a illustrates the nature of this interaction effect, showing that in schools with a high number of repeaters, repeaters tend to present a higher perception of reading competence than promoted students.

### 3.2.2. Effects of Grade Retention on Goal Orientations

Considering students' task orientation and self-enhancing orientation, the ICCs from Model 0 revealed that 19% and 16% of the variance in students' goal orientations is attributable to schools. The results from all models showed that retention is related to lower levels of task orientation (b = −0.12, $p < 0.001$; b = −0.11, $p < 0.001$; b = −0.16, $p < 0.001$; Models 1 to 3, respectively). Considering the effects of school retention composition, we found an effect of this variable only in Model 3 (b = −0.13, $p < 0.001$) when the interaction term was entered. The moderating effect of school retention composition in the relationship between individual retention and task orientation (b = 0.13, $p < 0.001$) showed that, on

the one hand, grade retention affects students' task orientation less when they attend a school with a higher number of retained students. On the other hand, an inspection of this interaction considering retained and non-retained students showed that for retained students, being in a school with high retention rates does not affect or even slightly improve their task orientation and, for promoted students, being part of such a school negatively affects their task orientation (see Figure 2b).

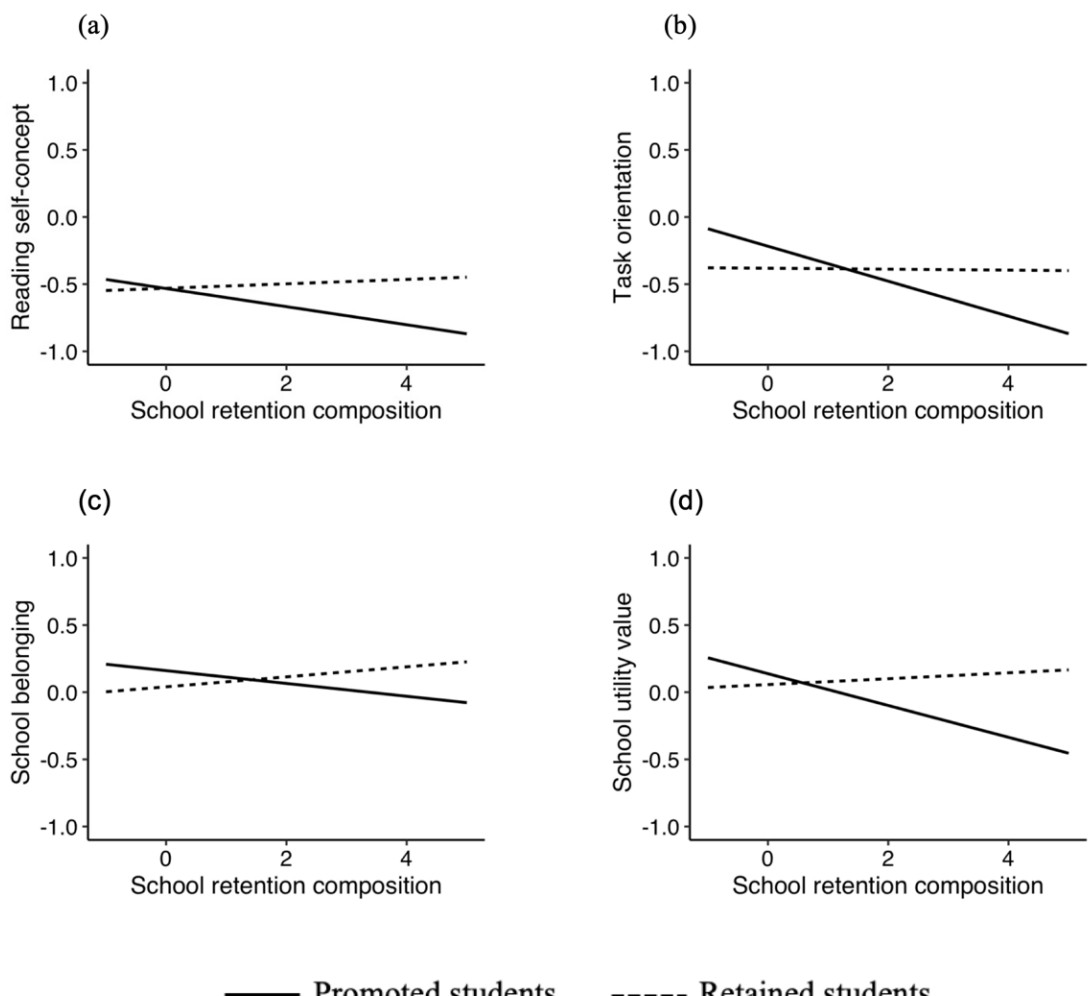

**Figure 2.** Graphical representation of the interaction effect between grade retention and school retention composition that resulted from multilevel hierarchical models (Model 3). Note. (**a**) Reading self-concept; (**b**) task orientation; (**c**) school belonging; (**d**) school utility value.

### 3.2.3. Effects of Grade Retention on School Belonging and Valuing

For school belonging and utility value, the ICCs again showed substantial variability between schools in these components (ICC = 0.23 and ICC = 0.18, respectively). The results for school belonging showed in both models that grade retention is related to lower feelings of school belonging (b = −0.09, $p$ = 0.004, for Models 1 and 2; b = −0.12, $p$ < 0.001, Model 3). In addition, consistent with the previous results, individual retention was found to be less detrimental to students' sense of belonging when schools present many repeaters (b = 0.09, $p$ = 0.001). Moreover, promoted students attending a school with a high rate of retained students present lower sense of school belonging (see Figure 2c). For students' sense of school utility value, this component was only affected by grade retention and school retention composition when the interaction term was considered (individual retention: b = −0.03, $p$ = 0.324, Model 1; −0.03, $p$ = 0.415, Model 2; b = −0.08, $p$ = 0.011,

Model 3; retention composition: b = −0.12, $p < 0.001$). Again, school retention composition was found to moderate the effects of individual retention (b = 0.14, $p < 0.001$), showing that grade retention rates at the school level tend to attenuate the negative relationship between grade retention and school utility value. Moreover, in line with previous findings, promoted students attending a school with a high rate of retained students present lower sense of school utility value (see Figure 2d).

## 4. Discussion

In Portugal, grade retention is still a prevalent practice applied to academically struggling students, irrespective of being considered an 'exceptional measure'. Therefore, based on data from PISA 2018, we aimed to explore the effects of grade retention on students' reading self-concept, goal orientations, and school belonging and valuing by employing methods that consider several background variables related to grade retention. The use of full matching [63] and the available data on several individual and contextual variables allowed us to reduce the differences between promoted and retained students and to estimate more rigorous effects of grade retention by disentangling the impact of potential confounders, as previously recommended [1,10,25].

### 4.1. Effects of Grade Retention and School Retention Composition on Psychosocial Outcomes

In this study, we found that retained students, by the age of 15, are less oriented to master academic-related tasks and have a lower sense of school belonging and valuing. These results are in line with previous studies, suggesting a detrimental effect of grade retention on motivational variables and school belonging [17–19,22,23,26,44]. Although the cross-sectional nature of this study limits our potential to make causal statements, our findings, together with the previous longitudinal and retrospective studies, suggest that retained students could engage in a negative cascade and that grade retention could leave an irreversible mark on students' motivation and engagement [15,16]. We additionally have explored the effects of repeating more than once by conducting separate analyses for students retained only once (n = 778) and students retained at least twice (n= 584). The analyses considering students retained only once revealed similar findings for all the outcomes. On the other hand, repeating more than once had positive effects on academic self-concept, negligible effects on task orientation and negative effects on students' self-enhancing orientation. The effects of school retention composition remained unaltered. Detailed results of these analyses are available on request from the first author.

These results are supported by the social goals framework, e.g., [42], stressing that students are most likely to engage in a context that provides opportunities to meet the social goal of establishing personal relationships with teachers and peers. Thus, in the case of retained students, by losing their reference peer group and friends, they could experience feelings of not being accepted, respected, or valued within their school community. In addition, students' motivation to learn is also affected by the fulfilment of social goals [42]. In this study, this is visible in the low levels of task orientation and school utility value presented by retained students, suggesting that these students are less oriented to develop learning task-related skills and gradually devaluate school learning.

Regarding students' reading self-concept, our results seem to unravel a more complex picture than individual grade retention effects alone. For this variable, we have found a more substantial effect of school retention composition over and above the impact of individual retention on students' perception of competence. These findings could also be interpreted in light of the big-fish–little-pound effect [28], which posits the school context's prevalence in shaping students' self-concept.

Indeed, one of the most notable findings of this study is the impact that school retention composition exerts on students' self-concept, motivation, and engagement. Overall, the larger the share of retained students in a school, the lower the self-perception of competence in reading, task orientation, and sense of school valuing. In addition, the interaction effect between individual retention and school retention composition suggests that the retention

rates do not affect students equally. In the case of our study, non-retained students were those that presented more negative outcomes by attending a school with a large proportion of repeaters. Thus, the adverse effects of grade retention also have significant implications for classmates of retained students and the broader school community, supporting previous findings [21,22,45,48,64].

These findings seem to reflect one of the major concerns of educators and parents—that sharing a classroom with repeaters by disrupting classroom instruction negatively affects the academic outcomes of their non-retained peers [65]. Thus, interventions, such as ability grouping, may sound attractive to ensure non-retained students' academic success. However, as previous studies showed, educational systems that preconise tracking and ability grouping in classes present the most detrimental effects of grade retention on students' development [6,20]. Moreover, retention rates and individual retention are related to students' background and achievement, meaning that schools with a large proportion of repeaters are most often attended by socially disadvantaged and low-achieving students [5,24,65], and these are simultaneously the characteristics that put students in a more vulnerable position to being retained [50,54,56]. Therefore, the most vulnerable students are, simultaneously, more likely to be retained and to share a class/school with many repeaters [48], resulting in greater inequalities among these students [24]. Here, tracking and ability grouping will only exacerbate this effect and, consequently, student disparities. Finally, at a macro level, the PISA 2018 data revealed that countries and economies presenting higher grade retention rates generally showed lower levels of reading performance and lower levels of equity in education [6].

### 4.2. Limitations and Future Directions

The contributions of the findings from this study are not without their limitations. First, we must mention the cross-sectional and retrospective nature of the presented data that prevents us from investigating developmental trajectories regarding students' academic self-concept, goal orientations, sense of belonging and valuing, and establishing causal relationships. In the case of grade retention research considering Portuguese data, longitudinal studies regarding grade retention effectiveness would be very important given the limited number of studies within this context and the high retention rates. Further investigations assessing Portuguese retained students' developmental trajectories of academic and psychosocial aspects are needed to clarify the effectiveness of this practice.

Second, and linked with the previous limitation, it was impossible to disentangle which grade students were retained considering the data used in this study. Thus, we could not estimate the potential long-term effects of grade retention more precisely. Considering the specific grade when students were retained is important not only from a developmental perspective but also due to findings from previous studies, where grade retention effects differed according to the grade where they were retained, e.g., [66].

Although some studies do not support the claim that repeating early or later grades produce differential effects on outcomes, from a developmental perspective, it would be important to consider and provide such information when estimating grade retention effects.

Third, concerning the moderating effect of school retention composition, our operationalisation of this variable is limited because only a small number of students from each school participated in the PISA assessment, and non-identification of the participating schools prevents us from obtaining school retention rates from official records. Based on this, one should interpret our findings regarding school retention composition with caution. Given its notable contribution, we encourage researchers to consider this vital variable in future studies assessing grade retention effects.

Fourth, and finally, we assumed the broad definition of grade retention, considering it as a single and universal treatment. Additional interventions coupled with grade retention during the retention year and beyond, such as additional support and educational services provided to retained students, were not considered. Past research showed that positive effects could emerge when retention is coupled with other treatment sources [20]. Moreover,

we did not investigate the potential moderating effects of students' background characteristics. Some researchers suggest that students from certain subgroups or presenting specific features might benefit from grade retention [20,54]. Although these aspects were beyond the scope of this study, we encourage researchers to collect information about 'what happens' during the retention year and test the effects of grade retention according to some students' characteristics in future studies. These two considerations will be very informative either to research or to practice.

## 5. Conclusions

The findings of this study suggest that grade retention is not an effective practice. Specifically, our results indicate that grade retention is related to lower levels of motivation and engagement, two valuable conditions for school success. Moreover, it was reported that grade retention affects retained students and could also be detrimental to their peers attending the same school.

Based on our findings, we cannot support the use of grade retention as an effective intervention for struggling students. Instead, we first recommend early identification of at-risk students, monitoring their academic and psychosocial development, and providing additional support to avoid grade retention. Valbuena et al. [25] listed numerous evidence-based and cost-effective policies, practices, and interventions that are alternatives to retention, such as tutoring, summer schools, and multi-age grouping. Likewise, students' academic competencies and psychosocial development should also be considered when deciding to retain a student, not only school marks. Furthermore, in the case of retaining a student, both these competencies should be monitored and supported equally, ensuring that the 'second chance' given to the students will not be harmful to their academic and psychosocial growth. To the broader school community, we recommend monitoring and reducing school retention rates since they affect the whole student community. At last, we advise educators and policymakers to continuously consider the psychosocial components of learning when debating the effectiveness of grade retention and its related policies and norms.

**Author Contributions:** J.P.: Conceptualisation, methodology, formal analysis, investigation, data curation, writing—original draft preparation, visualisation, project administration, funding acquisition. F.P.: Conceptualisation, writing—review and editing, supervision, funding acquisition. All authors have read and agreed to the published version of the manuscript.

**Funding:** This research was funded by Portuguese national funds via Fundação para a Ciência e Tecnologia (FCT) under grant [SFRH/BD/132195/2017] awarded to the first author, and grants [UIDP/04853/2020] and [UIDB/04853/2020] awarded to CIE-ISPA.

**Institutional Review Board Statement:** Not applicable.

**Informed Consent Statement:** Not applicable.

**Data Availability Statement:** Publicly available datasets were analysed in this study. These data can be found here: https://www.oecd.org/pisa/data/2018database/ (accessed on 18 March 2021). Our computations of the data are available upon request from the first author.

**Acknowledgments:** The authors wish to thank Timo Van Canegem for the valuable comments and suggestions provided for the manuscript.

**Conflicts of Interest:** The authors declare no conflict of interest.

## Appendix A

**Table A1.** Multilevel unstandardised parameter estimates for reading self-concept.

|  | Model 0 | Model 1 | Model 2 | Model 3 |
|---|---|---|---|---|
| Intercept | −0.48 (0.02) *** | −0.11 (0.08) | −0.11 (0.08) | −0.10 (0.08) |
| Student level |  |  |  |  |
| Retention |  | 0.03 (0.02) | 0.04 (0.03) | 0.01 (0.03) |
| Age |  | 0.01 (0.01) | 0.01 (0.01) | 0.01 (0.01) |
| Male |  | −0.06 (0.02) ** | −0.06 (0.02) * | −0.06 (0.02) * |
| ESCS [a] |  | 0.07 (0.01) *** | 0.07 (0.01) *** | 0.08 (0.01) *** |
| Immigration status |  | 0.02 (0.04) | 0.02 (0.04) | 0.03 (0.04) |
| Home language |  | −0.24 (0.06) *** | −0.24 (0.06) *** | −0.26 (0.06) *** |
| HEDRES [b] |  | 0.10 (0.01) *** | 0.10 (0.01) *** | 0.10 (0.01) *** |
| EMOSUPP [c] |  | 0.06 (0.01) *** | 0.06 (0.01) *** | 0.07 (0.01) *** |
| Reading score |  | 0.36 (0.03) *** | 0.36 (0.03) *** | 0.37 (0.03) *** |
| Math score |  | −0.31 (0.03) *** | −0.31 (0.03) *** | −0.31 (0.03) *** |
| Science score |  | 0.12 (0.03) *** | 0.13 (0.03) *** | 0.12 (0.03) *** |
| School level |  |  |  |  |
| Retention composition |  |  | −0.01 (0.02) | −0.07 (0.03) * |
| Public school |  | 0.10 (0.07) | 0.11 (0.07) | 0.11 (0.07) |
| School ESCS [a] composition |  | −0.03 (0.02) | −0.04 (0.03) | −0.04 (0.03) |
| School immigrant composition |  | −0.01 (0.02) | −0.01 (0.02) | −0.01 (0.02) |
| Retention X retention composition |  |  |  | 0.08 (0.02) *** |
| Between school variance | 0.09 (0.29) | 0.08 (0.28) | 0.08 (0.28) | 0.08 (0.28) |
| Within school variance | 0.34 (0.59) | 0.31 (0.56) | 0.31 (0.56) | 0.31 (0.56) |
| $R^2$ | 0.29 | 0.20 | 0.20 | 0.21 |

Note: Standard errors are in parenthesis. [a] Index of economic, social, and cultural background; [b] Home educational resources; [c] Parental emotional support. Standard errors are in parenthesis. * $p < 0.05$, ** $p < 0.01$, *** $p < 0.001$.

**Table A2.** Multilevel unstandardised parameter estimates for task orientation.

|  | Model 0 | Model 1 | Model 2 | Model 3 |
|---|---|---|---|---|
| Intercept | −0.18 (0.03) *** | 0.11 (0.10) | 0.11 (0.10) | 0.12 (0.10) |
| Student level |  |  |  |  |
| Retention |  | −0.12 (0.03) *** | −0.11 (0.03) *** | −0.16 (0.03) *** |
| Age |  | 0.03 (0.01) | 0.02 (0.01) | 0.02 (0.01) |
| Male |  | −0.22 (0.03) *** | −0.22 (0.03) *** | −0.21 (0.03) *** |
| ESCS [a] |  | 0.02 (0.02) | 0.02 (0.02) | 0.02 (0.02) |
| Immigration status |  | 0.18 (0.05) ** | 0.18 (0.05) *** | 0.20 (0.05) *** |
| Home language |  | 0.07 (0.07) | 0.07 (0.07) | 0.05 (0.07) |
| HEDRES [b] |  | 0.16 (0.01) *** | 0.16 (0.01) *** | 0.16 (0.01) *** |
| EMOSUPP [c] |  | 0.20 (0.01) *** | 0.20 (0.01) *** | 0.21 (0.01) *** |
| Reading score |  | −0.12 (0.04) *** | −0.11 (0.04) *** | −0.10 (0.04) ** |
| Math score |  | −0.06 (0.04) | −0.06 (0.04) | −0.07 (0.04) |
| Science score |  | 0.20 (0.04) *** | 0.20 (0.04) *** | 0.20 (0.04) *** |
| School level |  |  |  |  |

**Table A2.** *Cont.*

|  | **Model 0** | **Model 1** | **Model 2** | **Model 3** |
|---|---|---|---|---|
| Retention composition |  |  | −0.04 (0.02) | −0.13 (0.04) *** |
| Public school |  | −0.09 (0.08) | −0.08 (0.08) | −0.07 (0.08) |
| School ESCS [a] composition |  | 0.01 (0.03) | −0.03 (0.03) | −0.04 (0.03) |
| School immigrant composition |  | −0.01 (0.02) | 0.01 (0.02) | 0.01 (0.02) |
| Retention X retention composition |  |  |  | 0.13 (0.03) *** |
| Between school variance | 0.12 (0.35) | 0.10 (0.32) | 0.10 (0.32) | 0.10 (0.32) |
| Within school variance | 0.52 (0.72) | 0.47 (0.68) | 0.47 (0.68) | 0.47 (0.68) |
| $R^2$ | 0.19 | 0.17 | 0.18 | 0.20 |

Note: Standard errors are in parenthesis. [a] Index of economic, social, and cultural background; [b] Home educational resources; [c] Parental emotional support. Standard errors are in parenthesis. ** $p < 0.01$, *** $p < 0.001$.

**Table A3.** Multilevel unstandardised parameter estimates for self-enhancing orientation.

|  | **Model 0** | **Model 1** | **Model 2** | **Model 3** |
|---|---|---|---|---|
| Intercept | −0.03 (0.03) | −0.06 (0.10) | −0.06 (0.10) | −0.05 (0.10) |
| Student level |  |  |  |  |
| Retention |  | −0.05 (0.03) | −0.05 (0.03) | −0.07 (0.03) |
| Age |  | 0.02 (0.01) | 0.02 (0.01) | 0.01 (0.01) |
| Male |  | 0.42 (0.03) *** | 0.42 (0.03) *** | 0.42 (0.03) *** |
| ESCS [a] |  | 0.02 (0.02) | 0.02 (0.02) | 0.02 (0.02) |
| Immigration status |  | −0.14 (0.06) * | −0.14 (0.06) * | −0.14 (0.06) * |
| Home language |  | −0.16 (0.07) * | −0.16 (0.07) * | −0.17 (0.07) * |
| HEDRES [b] |  | 0.04 (0.02) * | 0.04 (0.02) * | 0.04 (0.02) * |
| EMOSUPP [c] |  | 0.11 (0.01) *** | 0.11 (0.01) *** | 0.11 (0.01) *** |
| Reading score |  | −0.15 (0.04) *** | −0.15 (0.04) *** | −0.14 (0.04) *** |
| Math score |  | −0.21 (0.04) *** | −0.21 (0.04) *** | −0.21 (0.04) *** |
| Science score |  | 0.34 (0.04) *** | 0.34 (0.04) *** | 0.34 (0.04) *** |
| School level |  |  |  |  |
| Retention composition |  |  | −0.01 (0.03) | −0.01 (0.03) |
| Public school |  | −0.01 (0.08) | −0.01 (0.08) | −0.01 (0.08) |
| School ESCS [a] composition |  | −0.01 (0.03) | −0.01 (0.03) | −0.01 (0.03) |
| School immigrant composition |  | −0.01 (0.02) | −0.01 (0.02) | −0.01 (0.02) |
| Retention X retention composition |  |  |  | 0.03 (0.03) |
| Between school variance | 0.11 (0.33) | 0.10 (0.31) | 0.10 (0.31) | 0.10 (0.31) |
| Within school variance | 0.54 (0.74) | 0.51 (0.72) | 0.51 (0.72) | 0.51 (0.72) |
| $R^2$ | 0.24 | 0.12 | 0.12 | 0.12 |

Note: Standard errors are in parenthesis. [a] Index of economic, social, and cultural background; [b] Home educational resources; [c] Parental emotional support. Standard errors are in parenthesis. * $p < 0.05$, *** $p < 0.001$.

**Table A4.** Multilevel unstandardised parameter estimates for school belonging.

| | Model 0 | Model 1 | Model 2 | Model 3 |
|---|---|---|---|---|
| Intercept | 0.08 (0.03) ** | 0.44 (0.10) *** | 0.44 (0.10) *** | 0.45 (0.10) *** |
| **Student level** | | | | |
| Retention | | −0.09 (0.03) ** | −0.09 (0.03) ** | −0.12 (0.03) *** |
| Age | | −0.03 (0.01) * | −0.03 (0.01) * | −0.03 (0.01) ** |
| Male | | 0.20 (0.03) *** | 0.20 (0.03) *** | 0.20 (0.03) *** |
| ESCS [a] | | 0.06 (0.02) *** | 0.06 (0.02) *** | 0.06 (0.02) *** |
| Immigration status | | −0.19 (0.05) *** | −0.19 (0.05) *** | −0.18 (0.05) ** |
| Home language | | 0.03 (0.07) | 0.03 (0.07) | 0.01 (0.07) |
| HEDRES [b] | | 0.09 (0.01) *** | 0.09 (0.01) *** | 0.09 (0.01) *** |
| EMOSUPP [c] | | 0.22 (0.01) *** | 0.22 (0.01) *** | 0.23 (0.01) *** |
| Reading score | | 0.09 (0.03) ** | 0.08 (0.03) * | 0.10 (0.03) ** |
| Math score | | −0.30 (0.03) *** | −0.30 (0.03) *** | −0.30 (0.03) *** |
| Science score | | 0.29 (0.04) *** | 0.29 (0.04) *** | 0.29 (0.04) *** |
| **School level** | | | | |
| Retention composition | | | 0.01 (0.03) | −0.05 (0.03) |
| Public school | | −0.32 (0.07) *** | −0.32 (0.07) *** | −0.32 (0.08) *** |
| School ESCS [a] composition | | 0.02 (0.03) | 0.03 (0.03) | 0.02 (0.03) |
| School immigrant composition | | 0.02 (0.02) | 0.02 (0.02) | 0.02 (0.02) |
| Retention X retention composition | | | | 0.09 (0.03) ** |
| Between school variance | 0.14 (0.38) | 0.10 (0.32) | 0.10 (0.32) | 0.10 (0.32) |
| Within school variance | 0.47 (0.69) | 0.43 (0.65) | 0.43 (0.65) | 0.43 (0.65) |
| $R^2$ | 0.23 | 0.22 | 0.22 | 0.25 |

Note: Standard errors are in parenthesis. [a] Index of economic, social, and cultural background; [b] Home educational resources; [c] Parental emotional support. Standard errors are in parenthesis. * $p < 0.05$, ** $p < 0.01$, *** $p < 0.001$.

**Table A5.** Multilevel unstandardised parameter estimates for school utility value.

| | Model 0 | Model 1 | Model 2 | Model 3 |
|---|---|---|---|---|
| Intercept | 0.20 (0.03) *** | 0.47 (0.10) *** | 0.47 (0.10) *** | 0.49 (0.10) *** |
| **Student level** | | | | |
| Retention | | −0.03 (0.03) | −0.03 (0.03) | −0.08 (0.03) * |
| Age | | −0.06 (0.01) *** | −0.06 (0.01) *** | −0.06 (0.01) *** |
| Male | | −0.16 (0.03) *** | −0.16 (0.03) *** | −0.16 (0.03) *** |
| ESCS [a] | | 0.02 (0.02) | 0.02 (0.02) | 0.02 (0.02) |
| Immigration status | | 0.07 (0.05) | 0.07 (0.05) | 0.09 (0.05) |
| Home language | | 0.08 (0.07) | 0.08 (0.07) | 0.06 (0.07) |
| HEDRES [b] | | 0.11 (0.01) *** | 0.11 (0.01) *** | 0.11 (0.01) *** |
| EMOSUPP [c] | | 0.22 (0.01) *** | 0.22 (0.01) *** | 0.22 (0.01) *** |
| Reading score | | 0.22 (0.03) *** | 0.22 (0.03) *** | 0.24 (0.03) *** |
| Math score | | 0.03 (0.03) | 0.03 (0.03) | 0.03 (0.03) |
| Science score | | −0.10 (0.04) * | −0.10 (0.04) ** | −0.10 (0.04) ** |
| **School level** | | | | |
| Retention composition | | | −0.03 (0.03) | −0.12 (0.03) *** |
| Public school | | −0.05 (0.07) | −0.04 (0.07) | −0.03 (0.07) |

**Table A5.** *Cont.*

|  | Model 0 | Model 1 | Model 2 | Model 3 |
|---|---|---|---|---|
| School ESCS [a] composition |  | 0.04 (0.03) | 0.02 (0.03) | 0.01 (0.03) |
| School immigrant composition |  | 0.01 (0.02) | 0.01 (0.02) | 0.01 (0.02) |
| Retention X retention composition |  |  |  | 0.14 (0.03) *** |
| Between school variance | 0.11 (0.33) | 0.09 (0.30) | 0.09 (0.30) | 0.09 (0.30) |
| Within school variance | 0.49 (0.70) | 0.43 (0.66) | 0.43 (0.66) | 0.43 (0.66) |
| $R^2$ | 0.25 | 0.23 | 0.23 | 0.25 |

Note: Standard errors are in parenthesis. [a] Index of economic, social, and cultural background; [b] Home educational resources; [c] Parental emotional support. Standard errors are in parenthesis. * $p < 0.05$, ** $p < 0.01$, *** $p < 0.001$.

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
