# Peer review of "One Step Back or One Step Forward? Effects of Grade Retention and School Retention Composition on Portuguese Students’ Psychosocial Outcomes Using PISA 2018 Data"

_sustainability, doi:10.3390/su142416573_

Round 1
Reviewer 1 Report
The article has the following advantages:
-addresses the important problem of class repetition and its psycho-social effects;
-transparency and methodological clarity;
-indication of the current research contexts of the problem from the perspective of literature analysis;
-interesting results, their interpretation and research conclusions.
Author Response
We would like to thank Reviewer 1 for the kind words and positive feedback regarding our work. We are happy that Reviewer 1 found our work interesting, clear and relevant. This encourages us to continue working in this field of research as well as to improve this study. Although Reviewer 1 did not raise concerns regarding the previous version of this study, we forward the revision letter to inform regarding the modifications made in the current version of the manuscript. Once more, we are grateful for the feedback provided by Reviewer 1.

Reviewer 2 Report
I’m happy to review the article, “One Step Back or One Step Forward…” Overall, I really liked this article. I do have two methodological concerns. The first is that students with multiple retentions are not accounted for, and secondly, I’m not sold on the idea of including covariates in the model that was already match using those same covariates. Otherwise, I think this article is beneficial to the domain of grade retention.
Introduction
I must say, the introduction is cogent and well-written. I appreciate the diverse schools of thought and thorough citations. Really sets up the paper well. That said, I will note that the introduction section is rather long, and I would suggest eliminating some of the commentary in a few sections.
I’m not sure figure 1 is necessary, as it really only relates to one or two sentences.
Methods
One thing I’m concerned about is that multiple retentions per student was not controlled for. In other words, someone who was retained multiple times seems much more likely to have negative outcomes, as compared to those retained once. The analyses either needs a covariate that assesses the number of times retained, or more preferably, exclude students retained more than once.
I’m not sold on the idea of including covariates in the same model that has matched pairs derived from those same covariates. I’ve looked over the Stuart article the authors cite, but couldn’t find a clear statement that advises this. Can the authors point me to where this is statistically appropriate in the Stuart article?
Results
“Conversely, Model 3 revealed a negative relationship between school retention composition and reading self-concept (b = -0.13, p < .05).” Something is wrong here. Did you mean task orientation? If that’s wrong, the sentence after this needs to be deleted.
I’m assuming there’s not a table with the ICC’s that are referenced throughout the results? Can the authors just note after the first ICC that they’re not shown in tables? This helps avoid the reader scanning through the tables (unless they’re buried in there somewhere).
Discussion
Generally well-written and thorough.
“Although some studies do not support the claim that repeating early or later grades produce differential effects on outcomes…” This is not necessarily true (see Giano et al., 2022) and understates a critical element in grade retention.
“Additional interventions coupled with grade retention… were not considered” but then the authors claim, “The findings of this study suggest that grade retention, without additional intervention strategies, is not an effective practice.” This is confusing because on one hand the authors state they couldn’t control for interventions, yet claim their results are “without additional intervention strategies.” If you didn’t control for it, you can’t claim “with or without” in your discussion.
Author Response
We would like to thank Reviewer 2 for the thorough comments and suggestions as they helped us to improve our work. We are happy that Reviewer 2 found our work relevant. We have carefully Reviewer 2's comments and suggestions in the current version of the manuscript. Our responses are listed in the Revision Letter. We are grateful once more for the feedback provided by Reviewer 2 as it helped to improve substantially the current version of the manuscript.

Reviewer 3 Report
This is a study of very high quality, both in its writing and in the analysis carried out and conclusions drawn. Some minor areas for improvement are suggested below:
- Use "grade retention" or "grade repetition" consistently throughout the text and not interchanging the expressions (e.g., p. 6).
- The literature review section could be improved by discussing previous studies that have considered each of the variables analyzed here, such as "school belonging and valuing" or "reading self-concept". If such studies do not exist, it would be useful to indicate this, which would highlight the contribution of the present article.
- I have missed a discussion of the small variance explained by each model. Yes, some of the variables analyzed are statistically significant, which is not surprising given the sample size. But in most cases the variance explained is less than 1%. Could this be a type II error (false positive)?
Author Response
We would like to thank Reviewer 3 for the thorough comments and suggestions as they helped us to improve our work. We are happy that Reviewer 3 found our work relevant and of quality. We have made great efforts to respond to Reviewer 3's comments. Our responses are listed in the Revision letter. We are grateful once more for the feedback provided by Reviewer 3 as it helped to improve substantially the current version of the manuscript.
